# Comprehensive Review of EGCG Modification: Esterification Methods and Their Impacts on Biological Activities

**DOI:** 10.3390/foods13081232

**Published:** 2024-04-17

**Authors:** Yingjun Zhuang, Wei Quan, Xufeng Wang, Yunhui Cheng, Ye Jiao

**Affiliations:** 1School of Food Science and Bioengineering, Changsha University of Science & Technology, Changsha 410114, China; zyj@stu.csust.edu.cn (Y.Z.); xufengw@csust.edu.cn (X.W.); cyh@csust.edu.cn (Y.C.); 2College of Food Science and Technology, Hunan Agricultural University, Changsha 410128, China; reus_quan@hunau.edu.cn

**Keywords:** epigallocatechin gallate (EGCG), esterification, biological activities

## Abstract

Epigallocatechin gallate (EGCG), the key constituent of tea polyphenols, presents challenges in terms of its lipid solubility, stability, and bioavailability because of its polyhydroxy structure. Consequently, structural modifications are imperative to enhance its efficacy. This paper comprehensively reviews the esterification techniques applied to EGCG over the past two decades and their impacts on bioactivities. Both chemical and enzymatic esterification methods involve catalysts, solvents, and hydrophobic groups as critical factors. Although the chemical method is cost-efficient, it poses challenges in purification; on the other hand, the enzymatic approach offers improved selectivity and simplified purification processes. The biological functions of EGCG are inevitably influenced by the structural changes incurred through esterification. The antioxidant capacity of EGCG derivatives can be compromised under certain conditions by reducing hydroxyl groups, while enhancing lipid solubility and stability can strengthen their antiviral, antibacterial, and anticancer properties. Additionally, esterification broadens the utility of EGCG in food applications. This review provides critical insights into developing cost-effective and environmentally sustainable selective esterification methods, as well as emphasizes the elucidation of the bioactive mechanisms of EGCG derivatives to facilitate their widespread adoption in food processing, healthcare products, and pharmaceuticals.

## 1. Introduction

Tea polyphenols are important bioactive compounds found in tea. Epigallocatechin gallate (EGCG), the most abundant component of tea polyphenols, is derived from 2-phenylbenzopyran and comprises three fundamental ring nuclei (A, B, and C), along with the gallic group on the D-ring [1]. Because of its remarkable biological activities, such as antioxidant [2], antibacterial [3], anticancer [4], and antiviral [5,6], EGCG has garnered significant attention. The biological activities of EGCG are related to its numerous active phenolic hydroxyl groups [7,8]. However, the presence of eight phenolic hydroxyl groups in its molecular structure also poses challenges in the practical application of EGCG. For example, the polyhydroxy structure of EGCG exhibits good water solubility but reduces its fat solubility, limiting its use in fat-soluble systems [9]. Moreover, because of its poor lipid solubility, EGCG encounters challenges in traversing the lipid bilayer of cell membranes in vivo, thereby impeding its effective delivery to target sites and, consequently, diminishing its bioavailability [10]. Additionally, the susceptibility of EGCG to decomposition stems from its phenolic hydroxyl structure when exposed to external environmental factors, such as light, heat, and alkalinity. Meanwhile, physiological environmental factors, including pH and enzymes in the organism, exert an influence on its stability [11]. To address these limitations, extensive research has been conducted on modifications of EGCG.

The modifications of EGCG can currently be classified into two distinct groups based on the specific sites of modification within the molecular structure of EGCG. One group is carbonylation, where the acylation site is located on a carbon atom in the aromatic ring. The other group is oxyacylation, also known as esterification modification, where the acylation site is located in the phenolic hydroxyl group [12]. Carbonylation introduces acyl groups to the benzene ring of EGCG, resulting in relatively stable derivatives. This method preserves the phenolic hydroxyl structure of EGCG, thereby enhancing its lipophilicity and stability while retaining its inherent antioxidant capacity. However, the high cost of reagents and the carcinogenic nature of commonly used solvents, like nitrobenzene, restrict its application in food [13]. Nevertheless, oxyacylation involves utilizing anhydride/chlorination to attach fatty chain groups to the phenolic hydroxyl group of EGCG. This method is typically conducted at ambient or moderate temperatures, facilitating the easy manipulation of reaction conditions. By employing appropriate ratios, it is possible to generate EGCG esterification products with favorable solubility and exceptional antioxidant properties [14].

This article provides a comprehensive review of the esterification methods employed for EGCG, thoroughly examines the impacts of structural modifications on the bioactivities of EGCG, and its application in food matrices, thereby enhancing our understanding of the structure–activity relationships exhibited by EGCG ester derivatives. Additionally, this study proposes future research directions and development strategies in this field. Overall, this study offers valuable guidance for synthesizing EGCG ester derivatives and establishes a theoretical foundation as well as development prospects for their application (Figure 1). 

## 2. Methods for Esterification and Analysis

The phenolic hydroxyl groups of EGCG are the most characteristic reactive groups. Esterification is a technique that introduces non-polar aliphatic hydrocarbon chains to the molecular structure of EGCG, thereby enhancing its lipid solubility and stability. EGCG esterification products can be achieved through two primary methods, namely, chemical and enzymatic approaches (Figure 2) [16,17].

### 2.1. Methods for Esterification

#### 2.1.1. Chemical Modifications

Chemical modifications were earlier applied to the esterification modification of EGCG. In the chemical esterification process, EGCG and the acyl donor are heated in a solvent with alkaline reagents as catalysts for several hours. Subsequently, the crude reaction product is obtained through washing, extraction, and drying procedures. The overall reaction process is influenced by factors such as the reagent type, reactant ratio, reaction temperature, and reaction time [18]. Commonly used solvents include ethyl acetate and acetone, while pyridine, sodium acetate, and sodium bicarbonate serve as catalysts. Anhydride, acyl chloride, and carboxylic acid are frequently utilized as acyl donors with varying degrees of acylation capacity, ranked in descending order of strength as follows: acyl chloride > anhydride > carboxylic acid [19]. Among these compounds, anhydride and carboxylic acid are employed to introduce short-chain fatty acid carbon chains (C2-C4). The incorporation of long-chain fatty acid carbon chains (above C6) into EGCG molecules is challenging because of significant steric hindrance and other factors. Consequently, acyl chloride is usually used as the acyl donor [18]. The chemical synthesis of EGCG esterification products is summarized in Table 1, encompassing variations in the acyl donor amount, catalyst amount, reaction temperature, reaction time, and yield.

Kohri et al. [20] and Lam et al. [15] used pyridine as a catalyst to react EGCG with acetic anhydride under heated and unheated conditions and obtained high yields of fully acetylated EGCG (AcEGCG) products, 98% and 82%, respectively. Zhong et al. [21] employed stearoyl chloride, eicosapentaenoyl chloride, and docosahexaenoyl chloride as acyl donors, catalyzing the reaction with pyridine at 50 °C under stirring to produce 3′,5′,3″,5″-*O*-acyl-EGCG in yields of 56.9%, 42.7%, and 30.7%, correspondingly. Pyridine serves the purpose for scavenging the hydrogen chloride (HCl) generated during esterification from the reaction medium, thereby facilitating the progress of the reaction [26]. However, pyridine poses toxicity as an organic base and presents challenges in achieving complete removal during the purification process [26]. Therefore, researchers have explored alternative low-toxicity or non-toxic reagents, such as sodium acetate and sodium bicarbonate, as alkaline agents. For instance, sodium acetate was utilized as a catalyst in the esterification reaction between palmitoyl chloride and EGCG in an acetone system, resulting in the successful synthesis of 4′-*O*-palmitoyl-EGCG, with an impressive yield of 63.3% [14]. Meanwhile, the use of sodium bicarbonate as a catalyst resulted in an EGCG–palmitate yield of 53.5% [23]. The higher yields obtained with sodium acetate can be attributed to its greater solubility in organic solvents, which allows it to act more effectively as an acid binder and enhance the yield [18].

The chemical method for the esterification of EGCG is well established and exhibits a high conversion rate [27,28]. However, its lack of selectivity leads to the generation of numerous complex reaction products that are challenging to isolate to purify the desired target products. For example, the esterification of EGCG with acetic anhydride, catalyzed by pyridine, yields three esters. The crude products are further separated by thin-layer chromatography and preparative high-performance liquid chromatography [29]. To achieve a purer target product, group protection is commonly adopted. Lin et al. [30] synthesized 3-*O*-acyl-EGC from EGCG by protecting the phenolic hydroxyl group with *tert*-Butyldimethylsilyl (TBDMS). In this method, EGC penta-*O*-TBDMS was prepared from the protected EGCG through reduction and ester bond cleavage reactions, followed by reaction with fatty acids, catalyzed by dicyclohexylcarbodiimide and 4-dimethylamino pyridine. The product was de-silicified to produce 3-*O*-acyl EGC. Hong and Liu [7] discovered that the hydroxyl groups on different rings in (+)-catechin, including rings A, B, and C, can undergo selective acylation with lauroyl chloride under alkaline or acidic catalytic conditions, resulting in the formation of five distinct substituted products. The addition of pyridine and 4-dimethylamino pyridine to a tetrahydrofuran solution of (+)-catechin promoted the monoacylation of ring C. Sodium bicarbonate and dimethyltin dichloride created an alkaline reaction environment, effectively achieving the monoacylation of ring B. The monoacylation of ring A was realized by introducing dihydrofuran and *p*-toluenesulfonic acid to provide a weakly acidic environment, which involved the sequential protection and deprotection of hydroxyl groups in rings B and C. Nevertheless, this group protection method involves multiple reaction steps and an extended duration. Hence, controllable, directional, environmentally friendly, and safe synthesis methods for EGCG esterification products are promising avenues to explore for future research.

#### 2.1.2. Enzymatic Modifications

Enzymatic modifications involve transesterification between EGCG and other esters, catalyzed by enzymes, resulting in the production of EGCG derivatives [31]. This method offers several advantages, such as mild reaction conditions, a straightforward reaction process, low reagent toxicity, elevated regional selectivity, and enzyme separation achieved through simple filtration and membrane separation [32,33]. Commonly used enzymes include lipases, proteases, and acyltransferases [34]. Lipases are the preferred catalysts for the acylation modification of phenolic compounds because of their outstanding regional selectivities and catalytic efficiencies [35]. These lipases are mainly derived from *Candida*, *Pseudomonas*, *Trichoderma*, and *Rhizopus*. Nevertheless, the sources of the lipases influenced their regioselectivities and catalytic efficiencies, leading to variations in the structures and yields of the derivatives [36,37] (Table 2). 

Zhu et al. [38] utilized the Lipozyme RM IM (from *Rhizomucor miehei*) to catalyze the reaction between EGCG and vinyl ester, achieving the production of 5″-*O*-acyl-EGCG and 3″,5″-di-*O*-acyl-EGCG in yields of 51.1% and 30.2%, respectively. By contrast, Jiang et al. [32] employed the lipase DF “Amano” 15 (from *Burkholderia cepacia*) as the catalyst in the presence of EGCG and vinyl ester, resulting in a conversion rate of 65.2% for the formation of 3″,5″-2-*O*-acyl-EGCG. The prevalent use of vinyl ester as an acyl donor in enzymatic esterification can be attributed to the facile conversion of vinyl alcohol, generated during the reaction, to acetaldehyde. This allows for the easy elimination of acetaldehyde from the reaction system, thereby shifting the equilibrium toward product formation [40,43]. Additionally, the carbon chain length of the acyl donor significantly influences the synthesis of EGCG esterification products. Compared to vinyl caprylate and vinyl laurate, the application of vinyl acetate as the acyl donor demonstrates a higher conversion rate under lipase catalysis [39]. This phenomenon can be ascribed to the elongation of the acyl donor’s carbon chain, leading to an augmented steric hindrance that impacts the formation of intermediate products and, ultimately, diminishes the catalytic efficiency of the lipase [44].

Lipases exhibit varying specificity and stability with different reagents [45]. Commonly used organic solvents for flavonoid acylation include acetonitrile, isoacetone, and hexane [46]. In the acetonitrile system, lipases exhibiting elevated structural stability demonstrated increased solvent tolerance, enabling the sustained maintenance of elevated catalytic activity levels [47]. Although organic solvents can enhance synthesis in lipase-catalyzed reactions, it is crucial to address the toxicity and removal of these solvents [33]. To address this concern, novel environmentally friendly and recyclable ionic liquids (ILs) are utilized as alternative solvents because of their advantages, including low volatility and high thermal stability [48]. The esterification reaction catalyzed by lipase shows superior performance when conducted in an IL compared to an organic solvent, as evidenced by an enhanced reaction rate, increased product conversion, and improved enzyme stability [49]. Zhu et al. [39] dissolved EGCG and vinyl acetate in 1-butyl-3methylimidazolium tetrafluoroborate ([Bmim][BF_4_]), catalyzed by the lipase Novozym 435, and heated the mixture at 70 °C for 10 h, resulting in the production of 5″-*O*-acetyl-EGCG and 3″,5″-di-*O*-acetyl-EGCG in yields of 43.56% and 54.79%, respectively. EGCG can also undergo reactions with vinyl esters of varying chain lengths (C2-C12) in [Bmim][BF_4_], yielding 5″-*O*-acyl-EGCG and 3″,5″-di-*O*-acyl-EGCG [11]. Although ILs possess numerous advantages, it is crucial to consider their significant and substantial mass transfer resistances and elevated costs. Furthermore, when ILs are used as solvent systems, they are difficult to remove during the separation of polyphenol derivatives. Therefore, it is necessary to investigate new green solvents that can be easily removed during esterification reactions.

### 2.2. Methods for the Analysis of EGCG Esterification Products

The analytical methods for EGCG esterification products encompass high-performance liquid chromatography (HPLC), mass spectrometry (MS), nuclear magnetic resonance (NMR), and infrared (IR) spectroscopy. These methods are utilized for the separation, quantification, and structural determination of esterification products. HPLC is a prevalent method for quantitative analysis, while mass spectrometry and NMR offer the precise characterization of the esterification products’ structures, and IR spectroscopy serves to detect chemical reactions [50,51]. For example, Liu et al. [14] used FT-IR to verify the esterification reaction between EGCG and palmitoyl chloride and detected the chemical structure of the EGCG palmitoyl ester as 4′-*O*-palmitoyl-EGCG by HPLC-MS, ^1^H-NMR, and ^13^C-NMR, and finally determined its content as 60.1% by HPLC.

## 3. Biological Activities of EGCG Esterification Products

The reported activities of EGCG, such as antioxidant, antiviral, and antibacterial activities, are intricately associated with the structure of its polyphenolic hydroxyl group [21,52,53]. The modification of EGCG not only affects the physical characteristics but also significantly influences the biological activities through changes in the number of phenolic hydroxyl groups and the polarity of the acyl moiety [12,36,54,55]. The biological activities of EGCG esterification products are comprehensively presented in Table 3.

### 3.1. Antioxidant Activity

The phenolic hydroxyl structure in EGCG contributes hydrogen to the free radical (R∙) formed during oxidation, thereby inhibiting or slowing the initiation of the free-radical chain reaction, ultimately halting the oxidation process and giving EGCG powerful antioxidant properties [68]. Thus, the antioxidant efficacy of phenolic compounds is primarily linked to the quantity and positioning of hydroxyl groups [69], while esterification modification alters the antioxidant potential of EGCG. The esterification of EGCG could theoretically diminish its antioxidant activity by reducing the number of phenolic hydroxyl groups available for hydrogen donation. It was observed that the scavenging abilities for DPPH free radicals, ABTS^+^ free radicals, and hydroxyl free radicals of EGCG–laurate decreased compared with those of EGCG [40]. Meanwhile, the antioxidant capacity of EGCG–laurate gradually diminished with an increase in the number of substituents. The synthesized 4′-*O*-palmitoyl-EGCG was found to have a lower ABTS^+^-radical-scavenging capacity than EGCG, which can be attributed not only to the diminished number of phenolic hydroxyl groups but also to the spatial site-blocking effect caused by the introduction of fatty acyl groups [14]. Jiang et al. [32] evaluated the antioxidant capacities of EGCG and its stearyl ester derivatives (3″,5″-2-*O*-stearyl-EGCG) using DPPH radicals and ABTS^+^-radical-scavenging tests. Although the antioxidant potencies of the EGCG stearyl ester derivatives were lower than that of EGCG, they exceeded that of 2,6-di-*tert*-butyl-homocyclophenol (BHT). 

The antioxidant capacity of acylated EGCG is also influenced by the type and positioning of substituents. Long-chain polyunsaturated fatty acids, such as eicosapentaenoic acid (EPA) and docosahexaenoic acid (DHA), demonstrate potent antioxidant activities, effectively mitigating reactive oxygen species (ROSs) in vivo [70]. These long-chain polyunsaturated fatty acids, when used as acyl donors, may synergistically enhance EGCG’s antioxidant capacity. Specifically, EGCG-3’,5’,3″,5″-*O*-Stearic acid (SA), EPA, and DHA tetraesters exhibited superior DPPH-radical-scavenging abilities compared to EGCG [21]. The effects of the substituents’ positions on the antioxidant activities of the derivatives were evaluated using the ABTS^+^-radical-scavenging-capacity test [7]. The replacement of the hydroxyl group at the C4’ position in the B ring of the catechins led to the lowest antioxidant capacity, followed by 3′-OH. In the catechin B ring, 4′-OH plays a crucial role in scavenging ABTS^+^ free radicals, which may be because of the high reducibility of the *ortho*-phenolic hydroxyl group. In addition, Peng and Shahidi [41] found that despite a decrease in the DPPH-free-radical-scavenging abilities of the EGCG monoester and diester (C2-C18) compared to that of EGCG, the DPPH-free-radical-scavenging abilities of EGCG esterification products increased with higher substitution degrees and longer substituent chain lengths. This is possibly because of the enhanced lipophilicity of EGCG esterification products, resulting in a stronger affinity with DPPH free radicals. Consequently, the antioxidant capacities of EGCG esterification products are affected by many factors, though acylation theoretically leads to a decrease in their antioxidant activities through the reduction of phenolic hydroxyl groups.

It is noteworthy that research focusing on the antioxidant capacities of EGCG ester derivatives should account for the employed solvent system. The antioxidant capacity of 4′-*O*-palmitoyl-EGCG surpassed those of EGCG and tocopherol in edible lard [14]. In both soybean and corn oils, the acetylated derivatives exhibited significantly higher antioxidant capacities than EGCG. Moreover, diacetylated EGCG demonstrated a stronger antioxidant capacity compared to monoacetylated EGCG [38]. The enhancement in the lipid solubility facilitates a more uniform dispersion in the oil, positioning the EGCG ester derivatives in closer proximity to the interface of the produced free radicals [11]. Additionally, the impact of the acyl chain lengths on the antioxidant efficacies of EGCG derivatives was also examined [11]. In soybean oil, the enhancement in the lipophilicities of the EGCG derivatives led to improvements in their antioxidant capacities. However, a critical point was observed at C8, beyond which the antioxidant effect diminishes, giving rise to a discernible “cutoff effect”.

The utilization of emulsion systems is widely observed in various food and biological contexts. Zhong and Shahidi [56] found that the tetraester of EGCG (EGCG-SA, EGCG-EPA, and EGCG-DHA) has a stronger antioxidant capacity than EGCG in the emulsion system. In contrast, Peng and Shahidi [41] investigated the antioxidant activities of EGCG derivatives in a β-carotene/linoleic acid emulsion and found no significant difference in antioxidant efficacies between EGCG monoesters (C2-C18) and EGCG. However, for the diacylation of EGCG, the antioxidant activities of the derivatives having acyl chain lengths exceeding four carbons exhibited diminished efficacies compared to that of EGCG, and they decreased with increasing chain length. Although the antioxidant effects of EGCG monoesters and diesters (C2-C18) are not higher than those of EGCG in emulsion systems, they exhibit greater significance in biological systems.

In biological systems, EGCG monoesters and diesters (C2-C18) were found to have a greater inhibitory effect on copper-induced LDL (low-density lipoprotein) cholesterol oxidation than EGCG, possibly because of the enhanced lipophilicities of EGGC ester derivatives [41]. This increased lipophilicity enables them to exhibit a stronger affinity toward LDL emulsion particles, thereby effectively protecting LDL from free radical attacks. EGCG tetraesters (EGCG-SA, EGCG-EPA, and EGCG-DHA) were also more effective than EGCG in preventing LDL cholesterol oxidation. Additionally, they effectively inhibited hydroxyl- and peroxyradical-induced DNA breakage as well as UV-induced liposome oxidation. This heightened efficacy is possibly attributed to the higher affinity between the esterification products of EGCG and cell membranes [56]. Esterification not only alters the formation of intramolecular hydrogen bonds in EGCG and modifies its hydrogen supply capacity but also induces changes in molecular conformations through steric effects, thereby influencing its interaction with free radicals. These alterations could result in superior performances of EGCG esterification products in reacting with free radicals [63].

### 3.2. Antibacterial Activity

EGCG exhibits broad-spectrum bacteriostatic activity. The antibacterial mechanism of EGCG involves the binding of its phenolic hydroxyl group or benzene ring to lipid molecules on the bacterial cell membrane through hydrogen bonds or hydrophobic interactions, disrupting the integrity of the cell membrane and resulting in the leakage of materials inside the cell, eventually leading to cell death. In addition, EGCG can disrupt bacterial cell walls by generating hydrogen peroxide (H_2_O_2_) via autoxidation [71,72,73]. It has been reported that the antimicrobial activities of EGCG ester derivatives exceed that of EGCG because of the introduction of hydrophobic fatty acid chains through the phenolic hydroxyl group in EGCG. Lipophilic EGCG derivatives demonstrate improved affinities for cell membranes, facilitating efficient penetration through the phospholipid bilayer into the cell for functional exertion [54] (Figure 3). 

Matsumoto et al. [54] observed a gradual increase in the antibacterial activities of EGCG derivatives as the acyl chain length increased. Among them, EGCG–palmitate exhibited potent bactericidal effects against Gram-positive bacteria (GPB) and rapidly eradicated methicillin-resistant *Staphylococcus aureus*. A previous study reported that the inhibitory effect of EGCG on Gram-negative bacteria (GNB) is comparatively lower than that on GPB, which can be attributed to the presence of lipopolysaccharide layers in GNB cell walls [74]. The lipopolysaccharide layers act as protective barriers for bacterial cells, hindering the penetration of hydrophilic EGCG. In comparison, the inhibitory effect of EGCG octaacetate on *Escherichia coli* (GNB) was found to be twice as strong as that of EGCG, potentially because of the introduction of hydrophobic side chains by EGCG. This modification enabled it to interact with the lipopolysaccharide layer of GNB and, consequently, caused damage to bacterial cell membranes [57]. Kajiya et al. [75] discovered that catechin derivatives had a strong affinity for cell membranes when the alkyl chain length exceeded 3 carbon atoms, while ester derivatives with an alkyl chain length exceeding 5 carbon atoms exerted detrimental effects on liposome membranes. Therefore, the affinity between cellular liposome membranes and catechin-acylated derivatives with different lengths of carbon chains plays a crucial role in elucidating their antimicrobial mechanisms.

Infections caused by spore-forming bacteria pose elevated morbidity risks within the food industry and medical settings. EGCG esters can weaken the spore shell and destroy the structural integrity, thereby inhibiting spore germination [58]. Alcohol and EGCG–palmitate-based surface disinfectant formulations produce high sporicidal effects on a broad spectrum of bacterial spores. The swift inactivation of spores is likely attributed to EGCG–palmitate damaging the spore shell’s structure, facilitating alcohol penetration of the spore matrix, and resulting in the prompt and irreversible inactivation of intrinsic biomolecules [76,77]. It has also been reported that EGCG stearate exhibits an enhanced binding affinity to the phospholipid bilayer, thereby inhibiting biofilm formation by *Streptococcus mutans* and exerting antibacterial effects [71].

### 3.3. Antiviral Activity

EGCG has been shown to have broad-spectrum antiviral activity against various viruses, encompassing DNA viruses, such as herpes simplex virus (HSV) [78] and hepatitis B virus (HBV) [79], and RNA viruses, including hepatitis C virus (HCV) [80], porcine reproductive and respiratory virus (PRRS) [81], and influenza virus [82], as well as human coronaviruses [83]. The potential of EGCG to inhibit various viruses through a shared mechanism makes it a promising candidate for the development of antiviral drugs. For instance, EGCG interacts with the surface proteins of HSV and HCV, inhibiting their attachment and invasion processes [84,85]. Additionally, EGCG competes with heparan sulfate for binding to HSV and HCV, while also competing with sialic acid for binding to influenza viruses [86,87]. However, the limited solubility of hydrophilic EGCG in fat-soluble systems impedes its delivery to the target site, thereby affecting its bioavailability [88]. It has been reported that EGCG ester derivatives demonstrate greater antiviral activities compared to EGCG, potentially attributed to the incorporation of fatty acid chains into the phenolic hydroxyl group of EGCG. This introduction of fatty acids not only enhances the stability and lipid solubility of EGCG but also increases its affinity with viral cell membranes, leading to a substantial improvement in its overall bioavailability [25] (Figure 3).

Kaihatsu et al. [60] discovered that the inhibitory effect of EGCG–palmitate on avian influenza virus (H5N2) in chicken embryos was 44-fold higher compared to that of EGCG. This enhanced efficacy can be accounted for by the increased affinity of EGCG–palmitate for virus–cell membrane surface proteins, thereby interfering with the binding between the capsid and the cell surface receptor sialic acid, effectively inhibiting the attachment and invasion of influenza viruses. EGCG–palmitate also exhibits inhibitory effects on HSV. The increased lipophilicity of EGCG–palmitate enhances its affinity for the glycoprotein present on the viral envelope, thereby facilitating the formation of a complex and effectively impeding virus attachment to and invasion of host cells [61]. Similarly, EGCG stearate demonstrates inhibitory effects on HSV without impacting the cellular morphology, suggesting its potential use as a topical treatment to mitigate the spread of HSV infection [89,90]. Furthermore, EGCG ester derivatives also exhibit the ability to inhibit HCV by targeting a characteristic protease of the virus [91]. The strong inhibitory effect on this protease can be attributed to alterations in molecular spatial–structural characteristics and the lipophilicity of EGCG, leading to an enhanced binding affinity with the enzyme [63].

The antiviral activities of EGCG ester derivatives were found to be influenced by the chain length. Mori et al. [25] investigated the effects of the chain lengths in EGCG ester derivatives (EGCG–butyrate, EGCG–octanoate, EGCG–laurate, EGCG–palmitate, and EGCG–eicosanoyl) on influenza A virus A/PR8/34 (H1N1). The results revealed that the antiviral activities of EGCG ester derivatives were 24-fold higher than that of EGCG, potentially because of the introduction of fatty acid chains to enhance the lipophilicity of EGCG and increase its cell membrane permeability. The antiviral efficacies of EGCG ester derivatives demonstrated an initial increase followed by a subsequent decline, corresponding to chain length augmentation, with the alkyl chain possessing eight carbon atoms emerging as a critical threshold. 

### 3.4. Anticancer Activity

EGCG has been shown to have a wide range of anticancer activities, including human endometrial [92], colon [93], breast [94], and liver [95] cancers. The anticancer mechanisms of EGCG primarily involve the inhibition of proteasome activities, induction of apoptosis, and suppression of tumor angiogenesis [4,96]. However, the presence of a phenolic hydroxyl group in the EGCG molecule gives rise to inherent limitations, such as poor lipid solubility, low stability, diminished bioavailability, and slow absorption in vivo [10,97]. The derivatives obtained through structural modification exhibited superior biological activities compared to natural EGCG in certain aspects. 

Matsumura et al. [98] investigated the anticancer activities of EGCG ester derivatives (EGCG–butyrate, EGCG–octanoate, and EGCG–palmitate), revealing a chain-length-dependent increase in their efficacies for suppressing tumor activities. The anticancer activities of EGCG–butyrate and EGCG–octanoate are related to their oxidation processes, leading to the generation of H_2_O_2_, which induces apoptosis [99]. In contrast, the longer fatty acid chain in EGCG–palmitate enhances its stability and inhibits H_2_O_2_ production. However, because of its high hydrophobicity, EGCG–palmitate can induce apoptosis by directly interacting with cell membranes and inhibiting the epidermal-growth-factor receptor (EGFR) activation. Chen et al. [42] reported the potent apoptosis-inducing ability of EGCG–laurate in human prostate cancer DU145 cells. This induction of apoptosis is primarily facilitated by the upregulation of the Bax/Bcl-2 ratio and the cleavage of cysteinyl asparaginase (Figure 3). Lam et al. [15] found that AcEGCG exhibited a stronger inhibitory effect on proteasome activity in leukemia cell lines compared to EGCG. Moreover, AcEGCG undergoes biotransformations in biological systems, involving acetyl group hydrolysis and subsequent degradation to EGCG, thereby extending the active group’s duration of action and enhancing its bioavailability [66]. An in vitro study using the human breast cancer cell line MDA-MB-231 revealed that AcEGCG exhibited a 2.8-fold higher rate of proteasome inhibition and a 2.1-fold stronger ability to induce apoptosis compared to EGCG [100]. Furthermore, the intracellular transformation and accumulation of AcEGCG were 2.4 times higher than those of retained EGCG. A mouse study also demonstrated that AcEGCG displayed a more potent capacity to inhibit the proliferation and metastasis of breast cancer cells compared to EGCG [100,101]. In addition, AcEGCG can interfere with angiogenesis by inhibiting signaling pathways, thereby suppressing angiogenesis in a human endometrial cancer model [64,65]. 

### 3.5. Other Biological Activities

The anti-inflammatory [102], antivitiligo [103], hypoglycemic [18], visual protective [104], and antiglycation [105] activities of EGCG esterification products have also been reported. The anti-inflammatory mechanism of EGCG primarily stems from its abilities to inhibit the production of pro-inflammatory factors and regulate the expressions of associated signaling-molecule genes. Specifically, the expressions of nuclear factor-κB (NF-κB), inducible nitric oxide synthase (iNOS), and cyclooxygenase-2 (COX-2) are inhibited by EGCG [106]. Zhong et al. [67] found that the esterification of EGCG with DHA increases its anti-inflammatory efficacy and suppresses the expressions of iNOS and COX-2 in murine colonic mucosa. The heightened anti-inflammatory activity of EGCG-DHA can be attributed to lipophilic DHA, which enhances its affinity with the cell membrane, thereby improving its cellular absorption capacity. Meanwhile, the hydrolysis of the EGCG-DHA ester results in the formation of free forms, potentially augmenting the anti-inflammatory efficacy through the synergistic interplay among these compounds [107,108]. Subsequently, the in vitro anti-inflammatory effects of EGCG–docosapentaenoic acid (EGCG-DPA) were evaluated using lipopolysaccharide (LPS)-stimulated mouse RAW264.7 macrophages as an inflammatory model. The EGCG-DPA ester exhibited superior anti-inflammatory activity compared to EGCG by suppressing the production of the pro-inflammatory mediators nitric oxide (NO) and prostaglandin E2 (PGE2) through the downregulation of iNOS and COX-2 gene expressions and inhibition of the activity of iNOS through polyphenol–protein interactions [102]. Additionally, the increased affinity of lipid-soluble EGCG-DPA toward NO free radicals enables the direct scavenging of NO, further contributing to its anti-inflammatory effects [109]. In conclusion, EGCG derivatives exhibit potential anti-inflammatory properties through the inhibition of inflammatory factors and modulation of inflammatory signaling pathways.

EGCG derivatives have potential in the prevention and treatment of vitiligo, diabetes, and eye diseases. The antivitiligo mechanism of EGCG involves the direct inhibition of the Janus kinase 2 (JAK2) activity and reduction in reactive oxygen species (ROSs) generated by oxidative stress [110]. Ning et al. [111] found that both AcEGCG and EGCG exhibit dose-dependent abilities to reduce intracellular ROS levels, suppress H_2_O_2_-induced melanocyte apoptosis, and inhibit the phosphorylation of p38 mitogen-activated protein kinase induced by H_2_O_2_. However, AcEGCG is more effective than EGCG in protecting melanocytes from oxidative damage, possibly because the phenolic hydroxyl group of EGCG is protected by the group to improve its stability. Additionally, AcEGCG effectively treated vitiligo by inhibiting the JAK1, JAK2, and JKA3 signaling pathways [103]. Consequently, AcEGCG can be regarded as a promising therapeutic agent for the treatment of vitiligo.

The antidiabetic activity of EGCG involves the inhibition of disease-related enzymes (such as α-glucosidase and α-amylase) to delay or slow carbohydrate digestion and absorption in the intestines, thereby enhancing blood glucose tolerance [112]. However, EGCG exhibits instability in an alkaline intestinal environment and exhibits diminished bioavailability because of its limited cell membrane permeability [113]. The introduction of fatty acyl chloride can improve the stability and bioavailability of EGCG [66,100]. Liu et al. [18] found that under alkaline conditions, EGCG was completely degraded after 5 h. Conversely, a significant amount of EGCG–palmitate hydrolyzed to EGCG after 1 h, and it took 5 h for EGCG–palmitate to be completely hydrolyzed and then oxidized to EGCG dimers, with no significant degradation products observed. Therefore, EGCG–palmitate enhances the stability of EGCG and, thus, prolongs its effective action time. Furthermore, the inhibitory effects of EGCG–palmitate on α-glucosidase and α-amylase were 52 times and 4.5 times higher than those of EGCG, respectively. 

Vascular endothelial growth factor (VEGF), a significant contributor to vision loss in the elderly, plays a pivotal role in the development of choroidal neovascularization (CNV) [114]. EGCG exhibits antiangiogenic effects by suppressing the expressions of inflammatory factors and VEGF in cells associated with diabetic retinopathy [115]. AcEGCG has been reported to have antiangiogenic effects as well [66], and it shows a decelerating effect on CNV-induced eye diseases, such as age-related macular degeneration [65,116]. Xu et al. [117] found significant efficacies of AcEGCG in diminishing vascular endothelial cell proliferation and migration and blood vessel formation. These effects were attributed to the downregulation of the HIF-1α/VEGF/VEGFR2 pathway and the modulation of M1-type macrophage/microglial cell polarization, ultimately leading to a significant decrease in CNV formation in mouse eyes. Similarly, Du et al. [104] revealed that AcEGCG reduced the production of proangiogenic factors induced by high glucose, achieved through the inhibition of the ROS/TXNIP/NLRP3 inflammasome axis in retinal Müller cells. Hence, AcEGCG shows promise in protecting eyesight through its antiangiogenic and stabilizing properties. 

Protein glycosylation triggers reactions linked to diabetic complications [118]. EGCG, a natural antiglycation agent, shows significant potential in mitigating the risk of diabetes [119]. Wang et al. [105] observed that introducing saturated fatty acids to EGCG does not affect its antiglycation activity, whereas the introduction of long-chain unsaturated fatty acids leads to a reduction in the antiglycation activity of EGCG because of steric hindrance effects.

Currently, research primarily focuses on exploring the antioxidant capacities of EGCG derivatives, encompassing various aspects, such as their potentials in anticancer, antiviral, anti-inflammatory, and antibacterial activities. However, it is crucial to further investigate other potential biological properties.

## 4. Application of EGCG Derivatives

Lipid oxidation is the primary cause of food spoilage [120]. However, certain synthetic antioxidants, such as butyl hydroxyanisole (BHA), dibutylhydroxytoluene (BHT), and terbutylhydroquinone (TBHQ), have been subjected to restrictions or bans in some countries because of their potential carcinogenic effects at high doses [121]. Conversely, EGCG is widely employed as a natural antioxidant in the food industry [9]. Nevertheless, the limited solubility of EGCG in lipid-based products, like oils and fats, significantly hampers its application as a food additive. Studies indicate that structural modifications to EGCG not only enhance its stability but also improve its overall performance in food systems [11,14,39,41].

### 4.1. Antioxidants

Lipid-based foods are susceptible to various oxidative reactions (photosensitive oxidation, auto-oxidation, and enzymatic oxidation) when exposed to light, oxygen, and water. These reactions can result in the degradation of food quality and the formation of hazardous substances, posing a significant challenge to food preservation and quality control [122]. Incorporating antioxidants into foods can hinder the initiation and progression of oxidative reactions by supplying electrons or hydrogen atoms to bind with free radicals generated during lipid oxidation [123]. The esterification products resulting from the acylation modification of EGCG exhibit excellent lipid solubilities and stabilities, presenting significantly enhanced antioxidant properties in lipid systems [12].

The solubility of EGCG is limited at 30 °C in lard, whereas EGCG–palmitate demonstrates a solubility of 0.04 g/100 g lard. Moreover, the solubility of EGCG–palmitate in lard at 90 °C surpasses that of EGCG by a factor of 470, resulting in enhanced oil transparency and color. The increased fat solubility of EGCG–palmitate enables an extension of the shelf life of lard to 8.4 months, representing a 3.5-fold increase compared to the original duration [14]. In addition, Liu et al. [23] found that the antioxidant efficacy of EGCG–palmitate in soybean oil is comparable to that of TBHQ and superior to that of EGCG. The addition of 200 mg/kg of EGCG–palmitate to soybean oil extended its shelf life to 16.61 months when stored at 25 °C. Currently, tea polyphenol–palmitate has obtained approval from the Chinese Food and Drug Administration for its application in food as an antioxidant for fats and oils [124].

### 4.2. Preserving Agents

Microbial intrusion during food processing and storage often leads to the problem of food spoilage. Adding preservatives to food is a critical approach to prolong its shelf life and ensure safety in consumption [125]. However, the excessive use of chemically synthesized preservatives, such as sodium benzoate, potassium sorbate, and nitrites, despite their cost-effectiveness and potent bacteriostatic effects, may pose significant threats to human health, including cancer induction, teratogenicity, and food poisoning [126]. Hence, the exploration of natural preservatives stands out as a focal point in current research efforts.

Zhou et al. [127] studied the effect of EGCG–palmitate on the preservation of fermented sausages, revealing its evident bacteriostatic effects, which significantly decreased the accumulation of intestinal flora during fermentation and, consequently, extended the shelf life of sausages. Moreover, the addition of EGCG–palmitate served a dual purpose by effectively preventing lipid oxidative rancidity and reducing the relative residual amount of nitrite. In addition, Shi [128] demonstrated that EGCG–palmitate was stronger than EGCG in inhibiting bacteria, such as *Streptococcus thermophilus*, during citrus storage while preserving the antioxidant properties of vitamin C through the maintenance of its content. 

## 5. Conclusions and Perspectives

The wide range of biological activities exhibited by EGCG has prompted its potential extensive utilization in the fields of food processing and the prevention and treatment of human diseases. Considering the inherent instability of EGCG, esterification represents an inevitable strategy for developing modified products with enhanced stabilities, heightened activities, reduced costs, and increased bioavailabilities. Simultaneously, esterification improves the lipophilicity of EGCG, thereby facilitating its application in fat-based food systems. Based on current research findings, future studies can be considered from the following aspects. First, it is essential to develop a cost-effective, environmentally friendly, and selective esterification method to promote in-depth research and facilitate the utilization of esterification products. Furthermore, investigating inherent mechanisms underlying various bioactivities of EGCG and its derivatives is imperative to establish a robust scientific foundation for devising innovative utilization strategies. Finally, enhancing safety assessments of EGCG derivatives is necessary to ensure their secure application in food processing as well as health care products or pharmaceuticals. In conclusion, research on EGCG esterification derivatives still requires further advancements to fully explore their potentials, as they hold promising prospects.

## Figures and Tables

**Figure 1 foods-13-01232-f001:**
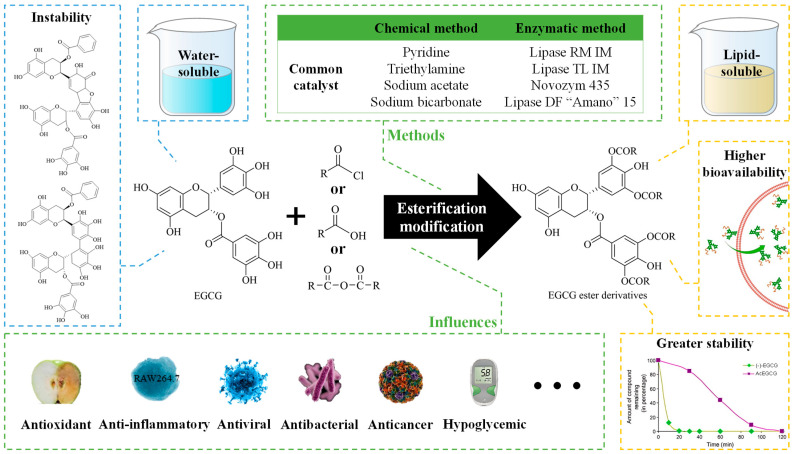
Graphical summary of the effects of esterification on the properties and biological activities of EGCG [15].

**Figure 2 foods-13-01232-f002:**
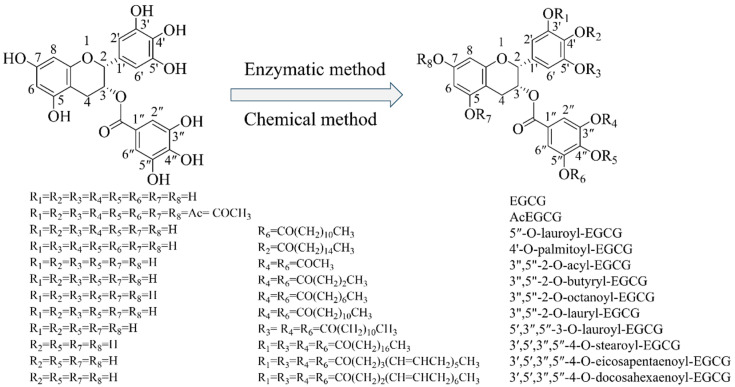
Structures of epigallocatechin gallate derivatives.

**Figure 3 foods-13-01232-f003:**
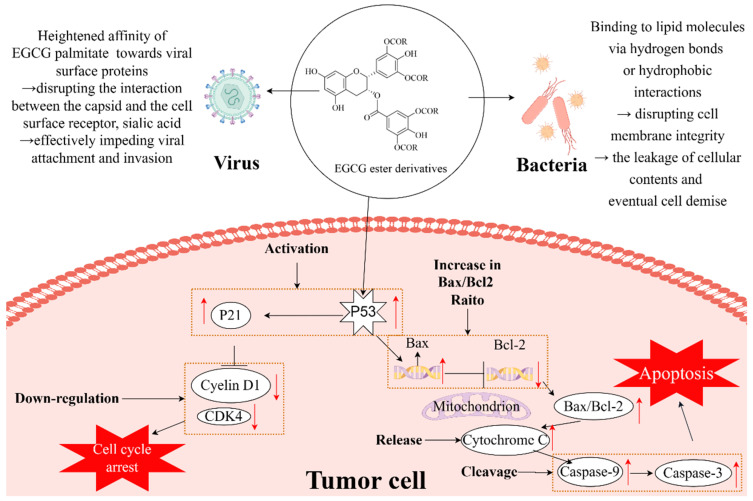
Anti-inflammatory, antiviral, and antibacterial mechanisms of EGCG ester derivatives. Graphic worked by Figdraw, red arrows represent the results of regulation (upward for upregulation, downward for downregulation), while black arrows indicate the orientation between steps of regulation or serve as an explanatory note for a specific step of regulation.

**Table 1 foods-13-01232-t001:** Chemical synthesis of EGCG esterification products.

Acyl Donor	Medium	Catalyst	Reaction Conditions		Yield (%)	Reference
			Temperature (°C)	Time (h)		
Acetic anhydride	-	Pyridine	45	20	98.3	[20]
Acetic anhydride	-	Pyridine	Room temperature	24	82	[15]
Stearyl chloride	Ethyl acetate	Pyridine	50	6	56.9	[21]
Eicosapentaenoyl chloride	Ethyl acetate	Pyridine	50	6	42.7	
Docosahexaenoyl chloride	Ethyl acetate	Pyridine	50	6	30.7	
Palmitoyl chloride	Ethyl acetate	-	40	3	46.3	[22]
Palmitoyl chloride	Ethyl acetate	Sodium bicarbonate	60	8	53.5	[23]
Palmitoyl chloride	Acetone	Sodium acetate	40	6	63.3	[14]
Oleyl chloride	Acetone	DMAP, DCC	Room temperature	12	98	[24]
Palmitoyl chloride	Tetrahydrofuran	Triethylamine	25	24	23	[25]

“-” denotes the absence of information. DMAP and DCC are 4-dimethylaminopyridine and dicyclohexylcarbodiimide, respectively.

**Table 2 foods-13-01232-t002:** Enzymatic synthesis of EGCG esterification products.

Acyl Donor	Medium	Lipase	Source of Lipase	Reaction Conditions		Yield (%)	Reference
				Temperature (°C)	Time (h)		
Vinyl acetate	Acetonitrile	Lipase RM IM	*Rhizomucor miehei*	40	8	84.5	[12]
Vinyl acetate	Acetonitrile/isopropyl alcohol	Lipase RM IM	*Rhizomucor miehei*	50	10	83.2	[38]
Vinyl acetate	[Bmim][BF_4_]	Novozym 435	*Candida antarctica lipase B*	70	10	98.7	[39]
Vinyl laurate	Acetone	Lipase DF “Amano” 15	*Rhizopus delemar*	50	96	80.1	[40]
Vinyl stearate	Acetonitrile	Lipase DF “Amano” 15	*Rhizopus delemar*	50	96	65.2	[32]
Vinyl fatty acids with different chain lengths (C2-C18)	*N*, *N*-Dimethylformamide	Lipase PS	*Burkholderia cepacia*	50	8	13.9~35.7	[41]
Vinyl fatty acids with different chain lengths (C4-C20)	*N*, *N*-Dimethylformamide	Lipase PL	*Pancreas*	57	1.5	35~39	[25]
Vinyl fatty acids with different chain lengths (C2-C12)	[Bmim][BF_4_]	Novozym 435	*Candida antarctica lipase B*	70	12	-	[11]
Lauric acid	Ethyl alcohol	Lipozyme TL IM	*Thermomyces lanuginosus*	45	12	63.2	[42]

“-” denotes the absence of information. [Bmim][BF_4_] is 1-butyl-3methylimidazolium tetrafluoroborate.

**Table 3 foods-13-01232-t003:** Comparison of biological activities of esterification products of EGCG and EGCG.

Bioactivity	Acyl Donor	Action Site	Model System	Effect	Reference
Antioxidant	C2	5′,3″,5″	DPPH; Sunflower oil	↑	[12]
	C12	5″; 3″,5″; 5′,3″,5″	Hydroxide; DPPH; ABTS Soybean oil	↓ ↑	[40]
	C16	4′	ABTS Edible lard	↓ ↑	[14]
	C16	4′	Sunflower oil	↑	[23]
	C2-C12	3″,5″	ABTS; DPPH; Hydroxide Sunflower oil; Oil-in-water emulsion	↓ ↑	[11]
	C2-C18	4′; 5′; 4″; 5″; 3′,5′; 3″,5″; 5′,5″; 4′,5″; 4″,5′; 4′,4″	DPPH; ABTS; FRAP Iron chelates	↓ ↑	[41]
	C18; C20; C22	3′,5′,3″,5″	DPPH	↑	[21]
	C18, C20, C22	3′,5′,3″,5″	Corn oil; β-carotene/linoleic acid; Fresh pork; LDL cholesterol	↑	[56]
Antimicrobial	C2	5,7,3′,4′,5″,3″,4″,5″	*Staphylococcus aureus*, *Bacillus subtilis* (GPB); *Escherichia coli*, *Yersinia enterocolitis* (GNB)	↑	[57]
	C16	3′,4′,4″,5″	*Staphylococcus aureus*, *Bacillus subtilis* (GPB); *Escherichia coli*, *Pseudomonas aeruginosa* (GNB)	↑	[54]
	C18	5′	*Bacillus cereus*, *Bacillus subtilis*	↑	[58]
	C18	5′	*Streptococcus mutans*	↑	[59]
Antiviral	C16	4′	Chicken eggs (influenza virus)	↑	[60]
	C16	4′	Vero cell (HSV-1)	↑	[61]
	C18	4′	Lung cell lines A549 and MRC-5 (Enterovirus 69)	↑	[62]
	C4-C20	3′; 4′; 4″; 5″	MDCK cells (influenza A/PR8/34)	↑	[25]
	C18; C20; C22	3′,5′,3″,5″	HCV (protease)	↑	[63]
Anticancer	C2	5,7,3′,4′,5″,3″,4″,5″	Endometrial cells	↑	[64]
	C2	5,7,3′,4′,5″,3″,4″,5″	AN3CA and RL95-2 (Human endometrial Cancer cells)	↑	[65]
	C2	5,7,3′,4′,5″,3″,4″,5″	KYSE150 (Human esophageal squamous cell) and HCT116 (Human colon cancer cell)	↑	[66]
	C12	3′,5′,3″,5″	DU145 cells and RWPE-1 cells (human prostate cells)	↑	[42]
	C22	3′,5′,3″,5″	Mice	↑	[67]

“↑” denotes heightened activity, while “↓” signifies diminished activity. DPPH, ABTS, and FRAP are 2,2-diphenyl-1-picrylhydrazyl, 2,2′-azino-bis (3-ethylbenzothiazoline-6-sulfonic acid), and ferric reducing antioxidant power, respectively. LDL is low-density lipoprotein. GPB and GNB are Gram-positive bacteria and Gram-negative bacteria, respectively. HCV and HSV are hepatitis C virus and herpes simplex virus, respectively.

## Data Availability

No new data were created or analyzed in this study. Data sharing is not applicable to this article.

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
