# Peer review of "Comprehensive Review of EGCG Modification: Esterification Methods and Their Impacts on Biological Activities"

_foods, 2024, doi:10.3390/foods13081232_

Round 1

Reviewer 1 Report

Comments and Suggestions for Authors

The manuscript foods-2955511 provides a narrative review of the esterification techniques applied to EGCG and their impacts on its bioactivity. This review contains interesting information and would be useful to the readership. it is well-written, well-structured, and nicely organized. I only found some suggestions for the authors to be addressed.

1. Line 13: the time period covered in this review related to the recent advances in this topic should be provided.

2. Line 71: A brief section covering the methods for chemical analysis of the esterification products can be added.

3. Line 279: Add a plausible explanation of the mechanisms (ET, HAT, combined) of EGCG derivatives for antioxidant capacity/activity.

4. Line 383: Specify the regioisomer of the AcEGCG. In this regard, a numbering sequence can be adopted to specify the regioisomers of the EGCG derivatives.

3. 

Author Response

Please note all revisions in the manuscript were marked in red font. The detailed responses to reviewer’s comments are as follows:

Point 1: Line 13: the time period covered in this review related to the recent advances in this topic should be provided.

Answer: We are grateful for the reviewer’s comments to improve our manuscript. We amend lines 11-13 of page 1 of the revised edition to read “This paper comprehensively reviews the esterification techniques applied to EGCG over the past two decades and their impacts on bioactivity.”

Point 2: Line 71: A brief section covering the methods for chemical analysis of the esterification products can be added.

Answer: Thanks for the reviewer’s comments. We have incorporated the analytical method for EGCG esterification products on page 6, lines 195-204, in response to your feedback. “The analytical methods for EGCG esterification products encompass high-performance liquid chromatography (HPLC), mass spectrometry (MS), nuclear magnetic resonance (NMR), and infrared spectroscopy (IR). These methods are utilized for separation, quantification, and structural determination of the esterification products. HPLC is a prevalent method for quantitative analysis, while mass spectrometry and NMR offer precise characterization of the esterification products' structures, and IR spectroscopy serves to detect chemical reactions47,48. For example, Liu et al.14 used FT-IR to verify the esterification reaction between EGCG and palmitoyl chloride, and detected the chemical structure of EGCG palmitoyl ester as 4′-O-palmitoyl-EGCG by HPLC-MS, 1H-NMR and 13C-NMR, and finally determined its content as 60.1% by HPLC.” Additionally, we have revised the headings of lines 71, 79, 80, and 143 to enhance clarity as follows: 2. Method for Esterification and Analysis; 2.1. Method for Esterification; 2.1.1 Chemical modification; 2.1.2 Enzymatic modification.

Point 3: Line 279: Add a plausible explanation of the mechanisms (ET, HAT, combined) of EGCG derivatives for antioxidant capacity/activity.

Answer: We thank the reviewer for this insightful comment. We have added “Esterification not only alters the formation of intramolecular hydrogen bonds in EGCG and modifies its hydrogen supply capacity but also induces changes in molecular conformation through steric effects, thereby influencing its interaction with free radicals. These alterations could result in superior performance of EGCG esterified products in reacting with free radicals [58].” in line 291-295, page 8-9 in the revised version.

Point 4: Line 383: Specify the regioisomer of the AcEGCG. In this regard, a numbering sequence can be adopted to specify the regioisomers of the EGCG derivatives.

Answer: Thank you for your review and valuable suggestions. Your highlighted the necessity to specify the regioisomer of AcEGCG. I wish to emphasize that the structure of AcEGCG has been thoroughly depicted in Figure 2, line 78, page 2. Considering this, redrawing the structure in revisions may appear redundant.

We would like to thank you again for your careful review of our paper and affirmation of our work. We hope that the revised manuscript has addressed all your concerns. Please let us know if there are other aspects that we need to work on further.

References:

  1. Liu, B.; Yan, W. Lipophilization of EGCG and effects on antioxidant activities. Food Chemistry 2019, 272, 663-669, doi:10.1016/j.foodchem.2018.08.086.
  2. Chen, P.; Du, Q.-Z. Isolation and Purification of a Novel Long-chain Acyl Catechin from Lipophilic Tea Polyphenols. Chinese Journal of Chemistry 2003, 21, 979-981, doi:10.1002/cjoc.20030210752.
  3. Wojtanowski, K.K.; Mroczek, T. Detection, identification and structural elucidation of flavonoids using liquid chromatography coupled to mass spectrometry. Current Organic Chemistry 2020, 24, 104-112.
  4. Zhong, Y.; Ma, C.-M.; Shahidi, F. Antioxidant and antiviral activities of lipophilic epigallocatechin gallate (EGCG) derivatives. Journal of Functional Foods 2012, 4, 87-93, doi:10.1016/j.jff.2011.08.003.

Reviewer 2 Report

Comments and Suggestions for Authors

Author Response

Responses to Reviewer 2:

Please note all revisions in the manuscript were marked in red font. The detailed responses to reviewer’s comments are as follows:

Point 1: the title and abstract are clear. but I prefer that the authors modify a few words of the title to clearly explain the objective of the study.

Answer: Thank the reviewers for their review and valuable suggestions. As per your request, I have revised the article title to “Comprehensive review of EGCG modification: esterification methods and impact on biological activity”. I am confident that this revision more accurately portrays the comprehensiveness and focus of the article's content, particularly in its systematic examination of EGCG esterification methods and their impact on biological activity.

Point 2: part: 2. Method for Esterification

2.1. Chemical modification

2.2. Enzymatic modification

is very clear in the figure and the table which summarizes the chemical and enzymatic synthesis of products esterified by EGCG.

Answer: We thank the reviewers for their kind words and positive feedback. I am grateful for your recognition and will continue to strive for improvement in my work.

Point 3: the part: 3.1. Antioxidant activity

Do you have references to justify the interpretation cited in the paragraph: for the diacylation of EGCG, the antioxidant activity of the derivatives having acyl chain lengths exceeding four carbons exhibited a diminished efficacy compared to EGCG, and it decreased with the increase in chain length. Although the antioxidant effects of EGCG monoesters and diesters (C2~C18) are not higher than those of EGCG in emulsion systems, they exhibit greater significance in biological systems.

Answer: Thank you to the reviewers for their questions! Yes, I have references to support my argument. Specifically, I refer to the following literature:

  1. Peng, H.; Shahidi, F. Enzymatic Synthesis and Antioxidant Activity of Mono- and Diacylated Epigallocatechin Gallate and Related By-Products. Journal of Agricultural and Food Chemistry 2022, 70, 9227-9242, doi:10.1021/acs.jafc.2c03086.
  2. Zhong, Y.; Shahidi, F. Lipophilised epigallocatechin gallate (EGCG) derivatives and their antioxidant potential in food and biological systems. Food Chemistry 2012, 131, 22-30, doi:10.1016/j.foodchem.2011.07.089.

These documents are cited in my thesis as [56], [57], and you can find the complete reference information in the literature list. I believe that these literatures provide a solid theoretical foundation for my research.

Point 4: Concerning the paragraphs of both parts: 3.2. Antibacterial activity and 3.4. Anticancer activity are acceptable in terms of interpretation and comparison of previous results.

Answer: We thank the reviewers for reviewing our thesis and for their positive comments. We will continue to monitor your other suggestions and are willing to further improve and refine the paper based on your guidance.

Point 5: For the party: 3.5. Other biological activities

The anti-inflammatory [97], anti-vitiligo [98], hypoglycemic [25], visual protective [99], and anti-glycation [100] 

I think it is necessary to detail each of the activities with these results found; even with small paragraphs because it is a review article.

Answer: We thank the reviewer for this insightful comment. We have added “Protein glycosylation triggers reactions linked to diabetic complications116. EGCG, a natural anti-glycation agent, shows significant potential in mitigating the risk of diabetes117. Wang et al.102 observed that introducing saturated fatty acids into EGCG does not affect its anti-glycation activity, whereas the introduction of long-chain unsaturated fatty acids leads to a reduction in the anti-glycation activity of EGCG due to steric hindrance effects.” in lines 476-481, page 12-13 in the revised version.

Point 6: For the parties: 4. Application of EGCG derivatives and 5. Summary and prospect are acceptable, just I prefer to modify, 5. Summary and prospect by 5. conclusion and perspectives

Answer: Thanks for the reviewer’s comments. Regarding your suggested revisions to Section 4, “Application of EGCG derivatives,” and Section 5, “Summary and prospect,” I will modify Section 5 to be titled “Conclusion and perspectives” to align more closely with academic conventions and stylistic preferences.

We would like to thank you again for your careful review of our paper and affirmation of our work. We hope that the revised manuscript has addressed all your concerns. Please let us know if there are other aspects that we need to work on further.

References:

  1. Wang, M.; Zhang, X.; Zhong, Y.J.; Perera, N.; Shahidi, F. Antiglycation activity of lipophilized epigallocatechin gallate (EGCG) derivatives. Food Chemistry 2016, 190, 1022-1026, doi:10.1016/j.foodchem.2015.06.033.
  2. Mazumder, K.; Biswas, B.; Al Mamun, A.; Billah, H.; Abid, A.; Sarkar, K.K.; Saha, B.; Azom, S.; Kerr, P.G. Investigations of AGEs’ inhibitory and nephroprotective potential of ursolic acid towards reduction of diabetic complications. Journal of Nat-ural Medicines 2022, 76, 490-503, doi:10.1007/s11418-021-01602-1.
  3. Wu, X.; Zhang, G.; Hu, X.; Pan, J.; Liao, Y.; Ding, H. Inhibitory effect of epicatechin gallate on protein glycation. Food Res Int 2019, 122, 230-240, doi:10.1016/j.foodres.2019.04.023.
